# Uncovering the Role of Hormones in Enhancing Antioxidant Defense Systems in Stressed Tomato (*Solanum lycopersicum*) Plants

**DOI:** 10.3390/plants12203648

**Published:** 2023-10-23

**Authors:** Paola Hernández-Carranza, Raúl Avila-Sosa, Obdulia Vera-López, Addí R. Navarro-Cruz, Héctor Ruíz-Espinosa, Irving I. Ruiz-López, Carlos E. Ochoa-Velasco

**Affiliations:** 1Facultad de Ciencias Químicas, Benemérita Universidad Autónoma de Puebla, Av. San Claudio y 18 Sur. Ciudad Universitaria, Puebla C.P. 72570, Mexico; paola.hernandezc@correo.buap.mx (P.H.-C.); raul.avila@correo.buap.mx (R.A.-S.);; 2Facultad de Ingeniería Química, Benemérita Universidad Autónoma de Puebla, Av. San Claudio y 18 Sur. Ciudad Universitaria, Puebla C.P. 72570, Mexico; hector.ruiz@correo.buap.mx (H.R.-E.); irving.ruiz@correo.buap.mx (I.I.R.-L.)

**Keywords:** tomato plants, secondary metabolism, antioxidants, phytohormones, reactive oxygen species

## Abstract

Tomato is one of the most important fruits worldwide. It is widely consumed due to its sensory and nutritional attributes. However, like many other industrial crops, it is affected by biotic and abiotic stress factors, reducing its metabolic and physiological processes. Tomato plants possess different mechanisms of stress responses in which hormones have a pivotal role. They are responsible for a complex signaling network, where the antioxidant system (enzymatic and non-enzymatic antioxidants) is crucial for avoiding the excessive damage caused by stress factors. In this sense, it seems that hormones such as ethylene, auxins, brassinosteroids, and salicylic, jasmonic, abscisic, and gibberellic acids, play important roles in increasing antioxidant system and reducing oxidative damage caused by different stressors. Although several studies have been conducted on the stress factors, hormones, and primary metabolites of tomato plants, the effect of endogenous and/or exogenous hormones on the secondary metabolism is still poorly studied, which is paramount for tomato growing management and secondary metabolites production. Thus, this review offers an updated overview of both endogenous biosynthesis and exogenous hormone application in the antioxidant system of tomato plants as a response to biotic and abiotic stress factors.

## 1. Introduction

Plants are generally subjected to biotic and abiotic stresses during their growth and development [1]. Among biological factors, microorganisms, insects, arachnids, and weeds are the most relevant, whereas radiation, salinity, floods, drought, extreme temperatures, and heavy metals are the most important abiotic stresses [2,3]. Although the genetic, molecular, and physiological mechanisms of plants against stress factors are not fully understood yet, available evidence indicates that these can cause negative, positive, or null effects on plants, and they cannot be generalized among species and stress factors [4]. Until now, it is well-known that, when plants are stressed, a complex signaling network is activated, initiating when the stress factor is perceived by protein receptors, triggering signal responses like reactive oxygen species (ROS) production, changes in basal levels of phytohormones, gene expression, kinase/phosphatase up- or downregulation, etc. [4,5].

ROS, such as hydrogen peroxide (H_2_O_2_), superoxide anion (O_2_^−^), singlet oxygen (^1^O_2_), hydroxyl ion (HO^−^), peroxyl ion (RO_2_^−^), alkoxyl ion (RO^−^), and organic hydroperoxide (ROOH), at basal levels, regulate plant growth and development. However, they are produced at higher levels under unfavorable conditions as a defense mechanism [6]. Their excessive production needs to be counteracted by plants to prevent oxidative damage and cell death through their antioxidant defense system [1]. Secondary metabolites (phenolic compounds, terpenoids, alkaloids, glucosinolates, etc.) and antioxidant enzymes (SODs, CATs, APXs, GPXs, etc.) are mainly responsible for managing ROS, maintaining cellular homeostasis, and reducing oxidative damage in plants [7].

Hormones play a crucial role in plant growth, organ formation, reproduction, fruiting, ripening, senescence, etc. In addition, one of the most important roles is protection against biotic and abiotic stress [8,9,10,11,12]. Each hormone initiates a specific pathway, which integrates a complex signaling network of synergistic, additive, or antagonistic interactions, commonly called crosstalk [13]. Hormones regulate key mechanisms, including up-or downregulation of gene transcription factors involved in encoding ROS production, enzymatic and non-enzymatic antioxidants biosynthesis, regulation of redox state, osmotic adjustment, physiological changes, hormonal homeostasis, etc. [14,15]. Therefore, understanding how biotic and abiotic factors affect the biosynthesis of hormones and their responses to maintain or even increase the antioxidant immunity system of plants is of great importance.

Tomato (*Solanum lycopersicum*) is one of the most important vegetable crops worldwide [16]. Its consumption is increasing due to its sensory attributes [17], its versatility [18], and its health-promoting compounds [19]. Despite tomato being one of the most studied plants, the effect of phytohormones on the antioxidant system (AS) against biotic and abiotic factors is scarcely reported. Thus, this review aimed to offer an overview of the hormone effects on improving the AS of tomato plants when they are subjected to stress factors. To fully achieve this purpose, the following topics are covered: (i) identify the enzymatic and non-enzymatic antioxidants in tomato plants and (ii) present and discuss the individual and crosstalk hormone responses on the AS of tomato plants subjected to stress factors.

## 2. Enzymatic and Non-Enzymatic Antioxidants in Tomato Plants

### 2.1. Non-Enzymatic Antioxidant Compounds (NEACs)

#### 2.1.1. Phenolic Compounds (PCs)

PCs are a diverse group of secondary metabolites having in common at least one benzene ring attached to one or more phenolic hydroxyl group substituents. PCs are generally classified into phenolic acids, flavonoids, xanthones, stilbenes, and lignans [20]. They are synthesized through the shikimic acid and phenylpropanoid pathways during normal growth or induced by biotic and abiotic stress factors [21]. PCs in plants can be electron or hydrogen donors to ROS molecules (^1^O_2_, H_2_O_2_, O_2_^−^, HO^−^, etc.). However, PCs also can activate the antioxidant enzyme system of plants. Like in many other plant products, the PCs composition of tomato fruit depends on several conditions, such as plant tissues, variety, growth and development condition, preharvest, harvest, and postharvest management, etc. Hydroxycinnamic acids (cis *ρ*-coumaric acid, 4-O-caffeoylquinic acid, gallic acid, ferulic acid, and chlorogenic acid) are the most abundant PCs reported in tomato plants [22,23]. Table 1 shows the main NEACs reported in tomato fruit.

#### 2.1.2. Carotenoids

Carotenoids are bioactive compounds belonging to the isoprenoid family; they provide yellow to red colors to fruits and vegetables [24]. Carotenoids are divided into two groups: carotenes, which contain only carbon and hydrogen in their molecular structure, and xanthophylls (XAN), which also possess oxygen in their different chemical forms [25]. All carotenoids present in fruits and vegetables are synthesized by the methylerythritol phosphate and mevalonate pathways, where, based on isopentenyl-diphosphate and dimethyl-diphosphate, lycopene is firstly synthesized, and then α-carotene or β-carotene are further produced [26]. Although carotenoids of tomato fruits are synthesized during the ripening process, on fully ripe tomatoes, lycopene (lyc) is the most abundant carotenoid, with 80–90%, and the rest of them are β-carotene, XAN, and other carotenoids [27]. Among carotenoids, lyc is the most efficient antioxidant compound because, according to its chemical structure (11 double bonds), it can act through the quenching of singlet oxygen and scavenging for peroxyl radicals [28].

#### 2.1.3. Vitamins

Among vitamins, E and C are the most potent antioxidants in fruits and vegetables [29]. Vitamin E is the common name given to the tocopherol (α, β, δ, and γ) and tocotrienol (α, β, δ, and γ) compounds, which are synthesized in the chloroplast by the tocochromanol pathway [30]. Among these compounds, α-tocopherol is the majoritarian tocopherol found in the mesocarp of tomato fruits, while γ-tocopherol is mainly found in seeds [31]. α-tocopherol is synthesized in the inner membrane of plastids and is responsible for protecting lipids and other membrane compounds of chloroplasts. When α-tocopherol scavenges ROS, it forms a tocopheroxyl radical, which is further reduced to tocopherol by the ascorbate-glutathione cycle [32]. Like many other phytonutrients, tocopherol levels change during plant growth and development, as well as in response to environmental factors such as light, temperature, nutrients, etc. [30,33].

On the other hand, vitamin C, or ascorbic acid (AA), is a potent water-soluble antioxidant compound of six-carbon lactone. It is a reducing agent; therefore, it can give electrons to any free radical, changing from ascorbic acid to semidehydroascorbic acid or dehydroascorbic acids, which are more stable compounds with a short life as free radicals [34]. Even though the tomato is not the best source of AA among fruits and vegetables, its higher consumption positions it as the most important for the human diet [35]. In this sense, the AA content in tomatoes is affected by several factors, like ripening; some authors have indicated that, as the tomato reaches edible maturity, the AA increases [36]. However, some other authors have pointed out that AA increases during the ripening of tomatoes, but when the fruit is fully ripe, the AA content significantly decreases [37].

#### 2.1.4. Glutathione (GSH)

GSH (γ-glutamyl-cysteinyl-glycine) is a key secondary metabolite for plant survival due to its role in ROS control [38]. Under normal conditions, it is commonly presented in reduced form (GSH), whereas its oxidized form (GSSH) is presented in a low amount [39]. In plants, the functions of GSH include storage of reduced sulfur, being a substrate for glutathione S-transferases for removing toxic compounds, maintaining the sulfhydryl groups of cysteine in their reduced form, and eliminating ROS generated by stress factors [39]. Contrary to the other non-enzymatic antioxidants, GSH has been less studied and quantified in tomato plants, but its concentration (Table 1) is also affected by several biotic and abiotic factors [40,41].

### 2.2. Antioxidant Enzymes (AEs)

#### 2.2.1. Superoxide Dismutases (SODs)

SODs are metalloenzymes responsible for catalyzing the dismutation of superoxide radicals (O_2_^−^) to O_2_ and H_2_O_2_ [42]. SODs are the principal defense against ROS, having an important role in treating oxidative stress diseases in living organisms. SODs are commonly grouped into four categories according to their metal cofactors: Cu/ZnSOD, FeSOD, MnSOD, and NiSOD [43]. However, only the first three have been widely found in lower and higher plants [44]. Cu/ZnSODs are mainly noticed in chloroplast, cytosol, and mitochondria, while FeSODs and MnSODs occur in chloroplasts and mitochondria, respectively [45]. In tomato plants, nine SOD genes (four Cu/ZnSODs, three FeSODs, and one MnSOD) have been unevenly distributed on 12 chromosomes [46]. Table 1 displays the AEs values reported in tomato plants.

#### 2.2.2. Catalases (CATs)

CATs are AEs presented in practically all living organisms, having vital roles in plant development and as a response to different stresses [47]. Among AEs, CATs were the first to be identified, and they are considered the most potent due to their affinity for the H_2_O_2_ radical (the major ROS), degrading to H_2_O and O_2_ [48]. CATs are unique enzymes because they do not require any cellular reducing equivalent as they mainly catalyze a dismutase retort [49]. They have been found in peroxisomes, mitochondria, cytosol, and chloroplast [50,51]. Although multiple CAT isoenzymes are reported in plants, in tomatoes, two isoforms (*CAT1* and *CAT2*) are related to stress factors [52,53,54].

#### 2.2.3. Ascorbate Peroxidases (APXs)

APXs are valuable components of the AEs of plants against biotic and abiotic stresses [55]. They are members of the class 1 peroxidases and have a vital role in the AsA-GSH cycle [56]. This cycle plays a principal role in plants. For example, they are used to detoxify H_2_O_2_ generated by the cytosol and chloroplast while maintaining ASA and GSH reserves in different cellular compartments [57]. APXs possess a higher affinity for reducing H_2_O_2_ by using ASA as an electron donor to H_2_O and MDHA, transforming the last compound to DHA [58]. Like many other AEs, APXs increase under various stress conditions [59]. In tomato plants, seven APX gene families (*APX1*, *APX2*, *APX3*, *APX4*, *APX5*, *APX6*, and *APX7*) have been found [60].

#### 2.2.4. Glutathione Peroxidases (GPXs)

GPXs are tetrameric enzymes containing seleno-cysteine (animal enzymes) or cysteine (plant enzymes) in their active site. GPXs catalyze the reduction of H_2_O_2_ to H_2_O or alcohols by the oxidation of GSH or thioredoxin [61]. Generally, plant GPXs prefer thioredoxin as a reducing agent instead of GSH [62]. In plant cells, these enzymes are localized in chloroplast, mitochondria, cytosol, and the endoplasmic reticulum [63]. Although different reports have indicated that GPXs are upregulated under various stress factors, many others have pointed out the downregulation of GPXs against stress factors [62,64]. In tomato plants, GPXle-1 is an isoform of GPX located in mitochondria and cytoplasm. This isoenzyme is associated with oxidative stress response [65]. However, Sharma et al. [66] pointed out that *SlPRX25*, *SlPRX75*, *SlPRX81*, and *SlPRX95* were upregulated in tomato plants infected with the tomato leaf curl New Delhi virus.

**Table 1 plants-12-03648-t001:** Reference values of AEs and NEACs of tomato fruit.

Component	Values	Source	Reference
Phenolic acids (mg/100 g DW ^1^)	172.19–311.82	Low ^3^	[22,67,68]
TFs (mg/100 g DW)	11.67–35.19	Low	[22,67,68]
Lyc (mg/100 g FW ^2^)	18.6–64.98	High	[27,69]
Total carotenoids (mg/100 g FW)	7.0–19.0	Medium	[27,68]
AA (mg/100 g FW)	16.32–19.43	Low	[36,68]
Tocopherols (mg/100 g FW)	0.17–0.62	Low	[31,68]
AC [FRAP (mmol/100 g DW)]	1.29–2.21	Medium	[68,70,71]
AC [DPPH (mmol/100 g DW)]	0.85–1.85	Low	[67,68,70]
GSH (mg/100 g FW)	1.43–1.61	NR ^4^	[72]
PODs (U/g FW)	7.03–19.8	NR	[73,74]
SODs (U/g FW)	0.35–0.65	NR	[73]
CATs (U/g FW)	2.08–26.91	NR	[73,74]
APXs (U/g FW)	10.25–14.05	NR	[73,74]

^1^ Dry weight; ^2^ Fresh weight; ^3^ The interpretation was conducted from comparative studies performed by the cited authors. ^4^ No reported, comparative studies were not found in the available literature.

## 3. Hormones and Their Effects on the Antioxidant System of Tomato Plants

### 3.1. Ethylene (ET)

Ethylene (C_2_H_4_) is a phytohormone associated with the ripening of climacteric fruits. At low concentration, it is responsible for tomato fruit’s color, taste, and flavor development [75]. Therefore, ET is involved in the biosynthesis of carotenoids (lyc and β-carotene), AA, TFs, PCs, and, consequently, the AC (FRAP and DPPH assays) of tomato fruit during the ripening [70,76]. Currently, Guo [77] has indicated that the histone deacetylation gene (SlHDT1) is a negative regulator of ethylene biosynthesis genes (*ACS2*, *ACS4*, *ACO1*, and *ACO3*), and it is vital for carotenoid gene expression and its accumulation. ET is also involved in plant growth, development, and stress response, and, due to its gaseous nature, can be easily transported in plants without a carrier [78]. Thus, ET plays an important role in accelerating the transition of primary to secondary metabolism when plants are stressed [79].

The effect of ET has been evaluated against biotic and abiotic stresses such as NaCl, CO_2_, CdCl_2_, microorganisms, cold temperature, etc., in seeds, plants, and/or tomato fruits, indicating that ET effects are time-, tissue-, dose-, and stressor-dependent [80,81]. Overall, results have indicated that, when tomato plants are stressed, ERFs, especially B1, E2, E3, F1, and F5, are upregulated by ROS production, increasing the AS [82,83]. However, some conflicting results are reported in the literature (Table 2). For example, tomato cells or plants treated with NaCl (0–250 mM) showed a higher positive correlation between ET accumulation and oxidative damage (EL, MI, or ROS) [84,85,86]. Nevertheless, in a recent study, tomato plants with over-expressing SlMAPK3 showed higher results of AEs (PRX, SOD, APX, and CAT) and lower values of OD than control and SlMAPK3 knock-out plants when NaCl (100 mM/L) was applied. The increase in AEs is attributed to the expression of genes related to ET biosynthesis [87]. Similarly, Gharbi et al. [88] indicated that *S. chilense* showed higher ET production compared to *S. lycopersicum* when they were stressed with NaCl (125 mM). The higher ET production was related to the increase in SlERF5, SlJERF1, and SlERF3 gene expression, causing low OD evaluated by MDA production. This study also indicated that tomato plants treated with AVG (aminoethoxyvinylglycine, ET inhibitor) had higher OD in leaf and root than untreated plants. Nevertheless, neither OD nor AEs was affected by ET in wild-type Micro-Tom and its Nr mutant when they were submitted to NaCl (100 mM) and CdCl_2_ (0.5 mM) as stressors [89]. These contradictory results may be explained since (1) tomato plants initiate the biosynthesis of ERFs in one tissue while suppressing its production in others; (2) ERFs are stress-dependent (one set for cold and heat, and another for salt, water, and flooding stresses), and (3) the possible activation of other hormones signaling pathways [83,87].

Regarding biotic stress factors, the information reported in the literature indicates more consistent results. Tian et al. [90] stated that ET reduces some transcription factors while increasing others in tomato plants against cotton ballworm (*Helocoverpa zea*) invasion, reducing PIN2 and PPOF gene expression, but inducing SA biosynthesis via ERF1 and PR1 upregulation. Overproducing ET tomato seeds (Micro-Tom variety) inoculated with mycorrhiza (*Glomus clarum*) produced higher values of transcription genes, such as CuZnSOD, CAT, and TPX1, related to AEs production [91]. Similarly, NEACs (AA and GSH) were increased in wild-type tomato seeds (LA0162) inoculated with *Bacillus megaterium* (PGPB) compared to its Nr counterpart, suggesting the intervention of ET in NEACs biosynthesis [92]. Recent studies have indicated that, when tomato plants were inoculated with *Fusarium oxysporum* or its toxin (fusaric acid), upregulation of ET key stress responses genes was observed (SlERF1, SlERF4, SlERF5, SlERF9, and SlERF11), which may reduce the OD in tomato plants [93,94].

The effect of ET in tomato fruit has also been evaluated during the cold storage (28–45 days at 7–8 °C). Results indicated that despite the ET increase being accompanied by an increment of lyc content, a significant reduction of AA, PCs, and AC was observed [95,96]. Similarly, when tomato fruits were stored for a long time period (35 days at 4 °C), higher ET and AEs were synthesized (associated with a higher expression of SlCBF1), and less OD (MDA and EL) were detected compared to antisense SlACS2 [14]. Therefore, an inverse correlation may exist between ET biosynthesis and OD, EAs, and NEACs. This tendency was corroborated by the evaluation of ET (dipped into 0.01% ethephon for 10 min) during the storage (2 °C for 20 h). The findings indicated a significant reduction of OD (MDA and EL) in treated tomato fruit [97]. Moreover, tomatoes at the breaker stage treated with ethephon solution (1.0 g/L, exogenous application) and stored at 25 °C increased their MDA, O_2_-, and H_2_O_2_ after 1, 2, and 6 days of storage, respectively [73]. Thus, it is possible to infer that ET treatment significantly improves the AS when tomato fruits are stressed, for instance, during low storage temperatures. On the contrary, ET treatment may induce OD [72,98].

**Table 2 plants-12-03648-t002:** Effect of ET on antioxidant system of tomato plants under biotic and abiotic stress.

Hormone (C ^1^)	Variety/Tomato Part	Age	Stress Condition	Results	Values	Reference
ET ^2^	Roma and Patio/microshoots	3 weeks old	NaCl (0–200 mM)	OD increases as ET increases.	EL = 23–80%	[84]
					MI = 0–73%	
					ET = 0.043–0.733 µ/Lh	
ET ^2^	Rio fuego/cells	4 to 5 days after subculture	NaCl (250 nM)	ROS increases as ET increases	H_2_DCFDA = 430–643%	[85]
					ET = 0.199–0.160 nL/g	
ET ^2^	Rio fuego/roots	6 weeks old	NaCl (100–250 mM)	ROS increases as ET increases	H_2_DCFDA = 103–133.4%	[86]
					EL = 117.90–428.66%	
					ET = 2.22–4.15 nL/g	
ET ^2^	Ailsa Craig and OE.MAPK3-5/roots	6 weeks old	NaCl (100 mM)	AEs and OD were increased in OE.MAPK3-5 tomato plants	POD = 103.58%	[87]
					SOD = 21.57%	
					APX = 11.34%	
					CAT = 48.90%	
					MDA = 39.02%	
					H_2_O_2_ = −48.6%	
ET ^2^	Ailsa Craig/roots and leaves	23-day-old seedlings	NaCl (125 mM)	Lower ET production in *S. lycopersicum* than in *S. chilense* produces higher OD	MDA = 36.15–59.67% ^4^; 36.07–57.86% ^5^	[88]
ET ^2^	Micro-Tom, epi, and Nr/roots	23 days old	*Glomus clarum* (10 g)	AEs increase in inoculated tomato plants	Cu/ZnSOD = 513.33%	[91]
					APX = 106.66%	
					CAT = 59.46%	
ET ^2^	WT and Nr/roots	8 weeks old	PGPB (10^−7^ CFU/mL)	PGPB increases NEACs in tomato plants	AA = 8.61–54.34%	[92]
					GSH = 24.28–37.90%	
ET ^2^	WT and Nr/leaves	6 to 7 weeks old	Fusaric acid (0.1–1.0 mM)	MDA was higher in Nr tomato plants and increased as FA increased	MDA = 2.17–55.4%	[94]
ET ^2^	Valouro/fruits	Ripe stage	Cold storage (7 ± 0.5 °C) for 35 days	Some NEACs were reduced as ET increased	ET = 51.74%	[96]
					AC = −16.2%	
					AA = −55.18%	
					PCs = −23.73%	
					Lyc = 92.0%	
ET ^2^	Calnegre/fruits	Breaker stage	Cold storage (8 ± 1 °C) for 28 days	After 28 days, ET and lyc increased, while AA was reduced.	ET = 57.35–268.9%	[95]
					Lyc = 22.8–42.4%	
					AA = −(54.9–63.4)%	
ET ^2^	WT and antisense SlACS2/fruits	Mature green stage	Cold storage (4 °C) for 35 days	MDA and EL were less in WT tomato compared to antisense SlACS2	ET = 564.1%	[14]
					MDA = −37.3%	
					EL = −31.7%	
ET ^3^ (0.01%)	Lichun/fruits	Mature green stage	Cold storage (2 ± 1 °C) for up to 3 weeks	Tomato fruit treated with ET presented less OD than untreated and 1-MCP treated	MDA = −(3.3–21.4)%	[97]
					EL = −(39.4–66.6)%	
ET ^3^ (100 µL/L)	BHN-602/fruits	Mature green stage	ET treatment and storage temperature (20 °C vs. 35 °C for 48 h)	ET treatment and higher temperature of storage increase NEACs	Lyc = 8.8%	[98]
					Carotenoids = 11.6%	
					PCs = 5.6%	
					AC (FRAP) = 13.8%	

^1^ Concentration; ^2^ Endogenous; ^3^ Exogenous; ^4^ Roots; ^5^ Leaves.

### 3.2. Salicylic Acid (SA)

SA is a natural phenolic compound (2-hydroxybenzoic acid) essential for signaling plant hormone immunity [99]. SA plays pivotal roles in plants in functions like stress tolerance, seed germination, DNA damage/repair, thermogenesis, increasing yield, etc. [100]. The exogenous application of SA induces the well-known SAR in different plant species against microbial pathogens such as viruses, fungi, and oomycetes, and some abiotic stress like salinity, drought, and heavy metals (Table 3) [101,102,103,104]. For example, when NaCl was applied as a stressor, exogenous SA application (0.1–1.0 mM) reduced EL [105], MDA [106], H_2_O_2_, and TBARS by increasing AEs such as SOD, GPX, APX, GR, and CAT [107,108]. The above-mentioned is probably caused by upregulating the expression of *HKT 1;2*, *NHX*, and *SOS1* genes, which regulate stress tolerance due to high salinity concentration [109,110]. 

During the growth of tomato plants at low temperature (10 °C), exogenous SA (0.5 and 1.0 mM) applied in two varieties (Streenb and Floridat) of tomato seedlings at 15 and 30 days after transplanting (one-true leaf stage) significantly increased AEs (PRX and PPO) and AC and reduced the OD indicators, such as MDA and EL, in leaf and root of tomato plants in a dose-dependent manner. It is worthy of interest that the H_2_O_2_ application displayed better AEs and OD responses than SA [111]. This result corroborates the idea that exogenous application or endogenous biosynthesis of ROS due to stress factors improves the AS of tomato plants. The effect of SA application during the tomato plant growth was evaluated against high-temperature exposure (42 °C for 36 h). The results indicated that SA increased AEs (SOD, CAT, PRX, APX), carotenoids, and proline and reduced EL, H_2_O_2_, and MDA [112]. They pointed out that SA supplementation improves the photosynthesis apparatus, proline production, and ROS management, important factors in avoiding OD. Moreover, the application of SA during tomato plant growth at low temperatures also improves the NEACs (AA and lyc) of tomato fruit [113]. The beneficial effect of SA in the tomato plant during its growth is time- and dose-dependent, showing an increase in AA, PCs, TFs, and AC as the dose of SA increases from 0 to 450 ppm; however, higher concentration was detrimental for tomato plants [114]. Moreover, some studies have indicated that, 24 h after SA application (0.2 and 1.0 mM), an increase in oxidative parameters is observed, which may trigger SAR [115,116]. For instance, SA application induces SAR in tomato plants against *Alternaria solani* [117,118], *Fusarium oxysporum* [119,120,121], *Xanthomonas vesicatoria* [122], *Potato virus X* [103], *Tomato yellow leaf curl virus*, *Orobanche* [123], *Ralstonia solanacearum* [124], *Tomato mottle mosaic virus* [125], and nematodes [126].

SA (0, 1, and 2 mM) was also applied to alleviate the effect of cold storage (cell membrane damage) on tomato fruit at the green mature stage. Results indicated that OD (MDA, EL, lipoxygenase, and phospholipase enzymes) was reduced probably due to the P5CS2 upregulation, which is responsible for proline production [127]. Moreover, SA application as a postharvest treatment of tomato fruit delayed the AA losses and lyc production during the storage at low temperature, which is associated with its antagonism effect on the ripening and senescence process caused by ET [128,129,130].

**Table 3 plants-12-03648-t003:** Effect of SA on antioxidant system of tomato plants under abiotic stress.

Hormone (C ^1^)	Variety/Tomato Part	Age	Stress Condition	Results	Values	Reference
SA ^2^ (0.1 mM)	Roma/roots	7-week-old	NaCl (150–200 mM)	SA application reduces OD	EL = −(32–44%)	[105]
SA ^2^ (0.01 mM)	Super Marmande/roots and leaves	35 days old	NaCl (100 mM)	SA application reduces OD	MDA = −(43.49–50.14)% ^3^ and−(23.62–25.88)% ^4^	[106]
SA ^2^ (0.1 mM)	Hezuo 903/leaves	47 days old	NaCl (100 mM) for 14 days	SA application improves the AEs and reduces OD	GSH = 60.1%	[107]
					H_2_O_2_ = −47.2%	
					TBARS = −53.9%	
					SOD = 31.6%	
					CAT = 41.5%	
					APX = 29.60%	
					GPX = −25.06%	
					DHAR = 76.0%	
SA ^2^ (1 mM)	Rio fuego/leaves	31 days old	NaCl (100 mM) for 7 days	SA application improves AEs	SOD = 46.8%	[108]
					CAT = 109.9%	
					APX = 494.9%	
					GR = 52.9%	
					AA = 29.5%	
					GSH = 52.6%	
SA ^2^ (100 mM)	Pusa ruby/leaves	75 days old	NaCl (250 mM) for 3 days	SA reduces the OD and increases the AEs	EL = −74.6%	[110]
					SOD = 158.8%	
					CAT = 137.3%	
					APX = 166.6%	
					GR = 172.7%	
SA ^2^ (0.5 and 1.0 mM)	Streenb and Floridat/leaves and roots	80 days old	Growth under low temperature (10 °C)	SA applied increases AS and reduces OD	POD = 7.5–42.2% ^3^; 15.8–34.0% ^4^	[111]
					PPO = 14.2–50.1% ^3^; 18.7–39.8% ^4^	
					AC = 21.4–31.6 % ^3^; 19.9–28.9% ^4^	
					MDA = −(13.6–33.3%) ^3^	
					EL = −(4.3–12.6%) ^3^	
SA ^2^ (200 ppm)	Super strain B/fruits	3 months old	Growth under changing temperatures (7.8–32.3 °C)	SA increases NEACs	AA = 20.6%	[113]
					Lyc = 8.4%	
SA ^2^ (1 mM)	Hezuo 903/leaves	8 days after, with leaves	Heat stress (42 °C for 36 h)	SA reduces OD and improves AS	EL = −27.8%	[112]
					H_2_O_2_ = −22.7%	
					MDA = −28.1%	
					SOD = 22.2%	
					CAT = 100.3%	
					APX = 32.1%	
					POD = 61.6%	
SA ^2^ (1 or 2 mM)	Newton/fruits	Mature green stage	Cold storage (1 °C) for 3 weeks	SA reduces OD	EL = −13.94%	[127]
					MDA = −2.2%	
					LOX = −(33.6–45.4)%	
SA ^2^ (4-mM foliar-applied plus 1-, 2-, or 4-mM by dipping 5 min)	Baraka/fruits	NP ^5^	Cold storage (10 °C) for 40 days	SA application reduces OD and increases AA, without the effect of the concentration used (1, 2, or 4 mM)	EL = −(46.6–48.0%)	[128]
					AA = 336.6–403.3%	
					APX = 447.6–455.5%	
SA ^2^ (0.2–1.2 mM)	Samrudhi/fruits	Mature (pink to light red color)	Cold storage /4–5 °C) for 21 days	As increased SA concentration, AA increases but reduces NEACs	AA = 17.9–58.3%	[129]
					Lyc = −(4.6–32.1)%	
					β-carotene = −(10.2–42.5)%	
SA ^2^ (0.5–2 mM)	Durinta/fruits	Pink maturity	5 or 20 °C for 20 days	As increased SA concentration reduces Lyc regardless of the storage temperature	Lyc = −(18.2–21.1)%	[130]

^1^ Concentration; ^2^ Exogenous; ^3^ Leaves; ^4^ Roots; ^5^ NP: Not provided.

### 3.3. Jasmonates

Jasmonic acid (JA) and its methyl ester (MeJA) exist naturally in a wide range of higher plants when they are stressed [131], triggering the hyperproduction of various secondary metabolites [132]. Both have been applied in seeds and leaves during plant growth and as a postharvest treatment against stress factors (Table 4). Tomato plants inoculated with nematodes and molds and treated with exogenous JA or MeJA showed higher expression of AEs (SOD, PRX, CAT, and GPX) and NEACs (AA, PCs, GSH, carotenoids, tocopherols, and some flavonoids) by upregulation of some antioxidant genes, such as *PAL5*, *C4H*, *CHS*, and *FLS*, which are related to the phenolic biosynthesis pathways (kaempferol, quercetin, xanthophylls, anthocyanins, and salicylic acid mainly) [10,133,134,135,136]. In this sense, despite microorganisms improving the AS of tomato plants by SAR, their application also increased the OD. Nevertheless, JA or MeJA application significantly improves AS and reduces OD to a level that is even less than in uninfected plants. Similar behavior was observed when salt was used as a stressing agent [131,137]. 

Moreover, when tomato plants were submitted to cold stress, MeJA application significantly reduced OD (MDA and EL) by increasing putrescine biosynthesis through upregulation of *ACD1*. It is important to highlight that, at room temperature, putrescine content did not change between treatments [138]. In this sense, SOD, POD, and GST (the enzyme responsible for reducing GSH) are the main enzymes responsible for avoiding the OD, while PCs were not affected, suggesting the use of other NEACs as an antioxidant against the OD [9,139], which was partially corroborated by Ding et al. [138].

Jasmonates were also applied to tomato fruit during off-vine ripening to evaluate their effects as postharvest treatment. Tomatoes stored under volatilized MeJA (44.8 µL/L) for 1 week (13 °C) showed an increase in AA, PCs, lyc, and β-carotene after 1 week of retail [140]. Moreover, at a lower temperature (2 °C), tomato fruit treated with MeJA (0.05 mM vapor phase 12 h at 20 °C) significantly increased AEs (SOD, PRX, CAT, and APX) and lyc content and reduced OD by increasing the expression of *SIMYC2* (helix–loop–helix transcription factor, which is the master regulator of JA mediated response) [141]. In the same way, tomato fruit at the mature green stage inoculated with gray mold (*Botrytis cinerea*) and treated with exogenous MeJA induces endogenous JA biosynthesis and ET production. The treatment caused an increase in enzymes (PAL, C4H, 4CL) related to the synthesis of PCs and POD (an enzyme responsible for inducing lignification of cell walls, reducing fungal invasion). Moreover, low relative expression of the SlMYC2 transcription factor was reported, which is interesting because, under some stress conditions such as cold and wound, it is promptly transcribed as a stress response mechanism. However, under pathogen attack, it is a negative regulator of JA response [142,143].

**Table 4 plants-12-03648-t004:** Effect of JA or MeJA on antioxidant system of tomato plants under biotic and abiotic stress.

Hormone (C ^1^)	Variety/Tomato Part	Age	Stress Condition	Results	Values	Reference
MeJA ^2^ (1 mM)	Beta/seedlings	15 days old	Microorganism (*Alternaria porri* f. sp. *Solani*)	MeJA application increases NEACs	PCs= 17.1–21.5%	[144]
					ATH= 12.6–14.1%	
JA ^2^ (0.01–100 nM)	Pusa Ruby/seedlings	7 days old	Microorganism (*Meloidogyne incognita*)	MeJA application reduces OD and increases AEs	O_2_^−^ = −(17.8–30.9)%	[136]
					SOD = 19.3–43.0%	
					POD = 15.4–48.6%	
					CAT = 14.5–52.5%	
					APX = 1.4–29.5%	
					DHAR = 18.8–51.9	
					GST = −(24.5–35.5)%	
					GR = 14.5–70.3%	
					PPO = −(10.9–43.8)%	
JA ^2^ (0.01–100 nM)	Pusa Ruby/seedlings	7 days old	Microorganism (*Meloidogyne incognita*)	MeJA application reduces OD and increases NEACs	H_2_O_2_ = −(15.2–40.7)%	[135]
					GSH = 18.8–63.1%	
					Carotenoids = 25.9–48.7%	
					TFs = 20.3–56.7%	
					ATH = 33.3–80.1%	
					XAN = −(8.8)−94.7%	
					AA = 7.9–28.9%	
					Tocopherols = 7.7–21.4%	
					PCs = 27.5–80.9%	
MeJA ^2^ (0–60 µM)	Rio Grande and Savera/leaves	50 days old	Seeds dipped into NaCl (5%) for 10 min	MeJA increases AEs	CAT = 6.0–30.2%	[131]
					PRX = 5.3–25.1%	
JA ^3^	Castlemart and its JA-deficient mutant/leaves	45 days old	CdCl_2_ (5–50 mg/kg soil)	WT tomato showed less OD and higher AEs compared to its JA-deficient mutant	MDA = −26.9%	[9]
					EL = −27.6%	
					H_2_O_2_ = −21.1%	
					SOD = 29.5%	
					POD = 28.9%	
					CAT = 243.6%	
JA ^2^ (1 nM)	NP ^4^/leaves	55 days old	NaCl (200 mM)	JA treatment reduces OD and increases AS	H_2_O_2_ = −35.2%	[137]
					MDA = −22.4%	
					AA = 40.3%	
					GSH = 8.6%	
					TF = 74.3%	
					SOD = 19.4%	
					CAT = 27.6%	
					APX = 20%	
					GR = 22.4%	
MeJA ^2^ (100 µM)	MicroTom/leaves	4-leaf stage	Cold stress (4 °C) for 24 h	MeJA increases putrescine and reduces OD	MDA = −41.6%	[138]
					EL = −19.8%	
MeJA ^2^ (44.8 µL/L)	Carousel/fruits	NP ^4^	Cold storage (13 °C)	After 2-weeks of storage (13 °C) MeJA improved NEACs	AA = 50%	[140]
					PCs = 87.4%	
					Lyc = 177.8%	
					β-carotene = 43.3%	
MeJA ^2^ (0.05 mM)	Badun/fruits	Mature green	Cold storage (2 °C) for 28 days	MeJA treatment showed less OD and higher AS than silencing MeJA	MDA = −(39.7–70.3)%	[141]
					SOD = 39.9–62.0%	
					POD = 47.7–63.6%	
					CAT = 36.6–54.6%	
					APX = 42.6–53.9%	
					Lyc = 24.1–51.9%	

^1^ Concentration; ^2^ Exogenous; ^3^ Endogenous; ^4^ NP: Not provided.

### 3.4. Abscisic Acid (ABA)

The plant hormone ABA has been associated with the ripening process of fruits and vegetables [79,145]. However, it is also a hormone for stress management in tomato seeds, plants, and fruits (Table 5). It is noteworthy that most of the studies about ABA effects on tomato plants have been conducted to understand how it affects the stress response against abiotic stress factors [116,146,147,148,149,150]. ABA improves stress resistance by closing the stomata of plant leaves, reducing the OD, and increasing the AEs (CAT, APX, and GR) in tomato roots and shoots. This response deals with the ROS imbalance caused by salt and drought stress [151]; meanwhile, Cd accumulation decreased in tomato plants due to ABA inhibiting iron-regulated transporter 1, which shows divalent cations like Cd. The results were corroborated when exogenous ABA application significantly alleviated the OD caused by salt [152], heat [8], and drought [153,154,155] by inducing the AsA-GSH cycle and promoting the AEs biosynthesis (SOD, POD, CAT, APX, and GR) [152]. Moreover, Zhou et al. [8,156] indicated that ABA application improves the RBOH1 transcription, which increases the stress resistance of the tomato plant against cold, heat, drought, and salinity. In this sense, a study conducted by Wang et al. [157] pointed out that exogenous ABA applications significantly influence ABA signaling pathway genes related to transcription factors (SlSnRK2, SlAREB, and SlPP2C) associated with biotic and abiotic stress responses [155].

**Table 5 plants-12-03648-t005:** Effect of ABA on antioxidant system of tomato plants under abiotic stress.

Hormone (C ^1^)	Variety/Tomato Part	Age	Stress Condition	Results	Values	Reference
ABA ^2^ (50 µM)	LA1698/leaves	Seedlings	NaCl (200 mM)	ABA application reduces OD and increases AS	MDA = −35.6%	[152]
					H_2_O_2_ = −29.6%	
					SOD = 3.1%	
					POD = 17.1%	
					CAT = 3.8%	
					GR = 138.7%	
					APX = −20.5%	
					AA = 40.4%	
					GSH = 5.8%	
ABA ^3^	Rheinlands/leaves	74 days old	NaCl (100 and 250 mM)	Silencing ABA mutants present lower NEACs	Carotenoids= −(60.6–74.6%)	[116]
ABA ^3^	Ailsa Craig/roots	13 days old	NaCl (150 mM)	Silencing ABA mutants reduce AEs and increase OD	APX = −33.9%	[150]
					CAT = −24.2%	
					MDA = 40.3%	
ABA ^2^ (50 µM)	PKM1/leaves	7 days after 4-fully-expanded-leaves stage.	Drought (7 days)	ABA application reduces OD and increases AE	H_2_O_2_ = −57.0%	[154]
					SOD = 10.2%	
					CAT = 233.3%	
					APX = 26.8%	
					GR = 6.0%	
ABA ^2^ (150 µM)	Micro-Tom/leaves	1 month	Drought (6 days)	ABA increases AEs compared to untreated plants	SOD = 6.2%	[155]
					CAT = 17.8%	
					APX = 32.0%	

^1^ Concentration; ^2^ Exogenous; ^3^ Endogenous.

### 3.5. Gibberellic Acid (GA)

GA is a diterpenoid carboxylic acid belonging to the gibberellin family. Among GAs, GA_3_ acts as a natural plant growth regulator against biotic and abiotic stress factors by preventing lipid peroxidation and regulating AS [158,159]. Results of exogenous GA application to tomato plants (Table 6) indicated that some NEACs (PCs, TFs, and GSH) and AEs (SOD, PPO, and APX) were upregulated in tomato plants stressed with NaCl [160,161]. Interestingly, carotenoids were not affected by the application of GA, probably due to both compounds being derived from geranylgeranyl diphosphate [146,161]. Regarding other abiotic stresses, such as cadmium or heat conditions, the results indicated that GA (0–10 µM, exogenous application) used in tomato plants (30-days-old) alleviates the damage caused by heavy metals such as Cd (0–20 µM). In this aspect, MDA was reduced, and CAT, GPX, and APX were increased in a dose-dependent manner [162]. In the same way, the application of GA (100 ppm) significantly improves CAT and PRX during treatments of tomato seedlings at low (10 °C) and high (45 °C) temperatures [163]. On the other hand, some reductions in metabolic (transpiration) and physiological (plant growth, stomatal closure, xylem vessel proliferation, and expansion) processes were observed in tomato GA biosynthesis deficient mutants exposed to drought conditions [164,165]. This response is a defense adaptation mechanism of plants to reduce the stress damage caused by the overproduction of H_2_O_2_ [166]. 

**Table 6 plants-12-03648-t006:** Effect of GA on antioxidant system of tomato plants under abiotic stress.

Hormone (C ^1^)	Variety/Tomato Part	Age	Stress Condition	Results	Values	Reference
GA ^2^ (0.4–0.6 mM)	BF1 and UC82B/leaves	45 days old	NaCl (200 mM)	GA improves AS	APX = 0–9.6%	[161]
					PPO = 15.1–16.0%	
					SOD = 32.1–59.2%	
					TFs = 18.8–100%	
					PCs = 10.7–19.1%	
					Carotenoids = 294.4–1980%	
GA ^2^ (100 µM)	NP ^4^/leaves	3 weeks old	NaCl (250 mM)	GA application improves AS	GSH = 99.6%	[160]
					MDA = −13.3%	
GA ^3^	Micro-Tom and procera mutant/shoots	30 days old	Drought (7 days)	GA production reduces MDA, but induces H_2_O_2_	MDA = −18.6%	[166]
					H_2_O_2_ = 41.1%	
GA ^2^ (10 µM)	CH/roots	60 days old	Cd (20 µM)	GA application reduces OD and increases AEs	CAT = 9.3%	[162]
					GPX = 20.9%	
					APX = 12.9%	
					MDA = −38.7%	
GA ^2^ (100 ppm)	Fayrouz, Aziza and N23-48/shoot	6 weeks old	Temperature of growth when tomato shoots were exposed to 10 and 45 °C	GA increases the AE	CAT = 1.5–13.9%	[163]
					APX = 9.2–56.7%	

^1^ Concentration; ^2^ Endogenous; ^3^ Exogenous; ^4^ Not provided.

### 3.6. Auxins

Auxins are phytohormones that play the most relevant role in plant development and growth [167]. Among auxins, IAA is the most detected auxin; however, naphthaleneacetic acid (NAA), indole-3-butyric acid (IBA), 4-chloroindole-3-acetic acid, and phenylacetic acid are also present in lower amounts [168]. Despite auxins being related to physiological processes, recent studies have indicated that they are implicated in responses against biotic and abiotic stress factors (Table 7). IAA has been applied to tomato seeds to evaluate their effects against Cd [169], salinity [170], heat stress [171], phytotoxins (benzoic and vanillic acids) [172,173], and parasite plants [123]. IAA improves the redox status of the plants by increasing AEs (SOD, CAT, PRX, APX, GPX, and those related to the AsA-GSH cycle) and NEACs (carotenoids, TFs, PC, tocopherols, and AC), and reducing some OD indicators like EL, MDA, and H_2_O_2_ production. It is interesting to note that IAA effects are dose-dependent because, at low concentrations (<5 µM), an increase in AS is reported, which is caused by an increase in H_2_O_2_ production [169,171,174,175]. However, when increasing IAA concentration, no effect of IAA was observed in unstressed plants, which was probably caused by a reduction in H_2_O_2_ biosynthesis [123,173].

**Table 7 plants-12-03648-t007:** Effect of auxins on antioxidant system of tomato plants under biotic and abiotic stress.

Hormone (C ^1^)	Variety/Tomato Part	Age	Stress condition	Results	Values	Reference
IAA ^2^ (50 µM)	Roots/cv. Navoday	30 days old	Cd (100 µM)	AS was improved and OD was reduced in IAA treated plants	Carotenoids = 8.1%	[169]
					APX = 28.0%	
					GR = 99.4%	
					AA = 99.3%	
					GSH = 133.3%	
					O_2_^−^ = −32.0%	
					H_2_O_2_ = −27.1%	
					MDA = 37.1%	
					EL = 52.6%	
IAA, NAA ^2^ (100 mg/L)	UC82B/leaves	NP ^3^	NaCl (200 mM)	IAA increases CAT activity	CAT= 274.1–311.1%	[170]
IAA ^2^ (50 nM)	Five Star F-1 hybrid/leaves	17 days old after seed germination	Heat shock (38 °C for 4 h)	IAA reduces OD and increases AE	MDA = −38.5%	[171]
					EL = 20.5%	
					CAT = 9.6%	
					POD = 7.7%	
					SOD = 16.5%	
IAA ^2^ (1 mM)	Pusa ruby/leaves	First fully expandedleaves	Benzoic acid (0.5–1 mM)	IAA reduces OD and increases AE	Carotenoids = 32.4–62.6%	[172]
					EL = −(21.4–30.4)%	
					MDA = −(19.7–28.3)%	
					SOD = 31.7–40.5%	
					CAT = 66.1–97.3%	
					APX = 54.1–57.7%	
					GPX = 45.8–50.7%	
IAA ^2^ (1 mM)	Pusa ruby/leaves	Fully expanded leaves	Vanillic acid (0.5–1 mM)	IAA reduces OD and increases AS	Carotenoids = 13.8–27.3%	[173]
					MDA = −(9.8–13.0)%	
					EL = −(31.6–55.3)%	
					H_2_O_2_= −(3.4–18.5)%	
					SOD = 21.6–28.0%	
					CAT = 21.4–28.9%	
					APX = 31.8–34.5%	
					GPX = 42.0–52.1%	
					PCs = 23.1–41.0%	
					ATH = 15.3–34.7%	
IAA ^2^ (0.09 mM)	NP ^4^/root and shoot	60 days old	*Orobanche ramose* L. infection	IAA application improves AS and reduces OD	AC = 131.2% 4; 80.0% ^5^	[123]
					PCs = 48.4% 4; 46.1% ^5^	
					TFs = 115.9% 4; 63.2% ^5^	
					Tocopherols = 40.6% ^4^; 40.6% ^5^	
					ASA = 21.7% ^4^; 20.6% ^5^	
					GSH = 27.5% ^4^; 168.8% ^5^	
					H_2_O_2_ = −8.1% ^4^; −26.2% ^5^	
					MDA = −17.1% ^4^; −24.5% ^5^	
					CAT = 37.2% ^4^; 31.1% ^5^	
					POX = 31.0% ^4^; 33.0% ^5^	
					SOD = 40.1% ^4^; 26.4% ^5^	
					APX = 30.4% ^4^; 66.8% ^5^	
					GR = 36.6% ^4^; 43.2% ^5^	

^1^ Concentration; ^2^ Exogenous; ^3^ Not provided; ^4^ Root; ^5^ Shoot.

### 3.7. Brassinosteroids (BRs)

BRs are a group of polyhydroxy steroidal phytohormones present in different parts of plants [176]. BRs participate in diverse physiological and developmental processes, such as growth, seed germination, rhizogenesis, senescence, and resistance against various abiotic and biotic stresses [177]. The exogenous application (Table 8) of BRs significantly increases the AEs (SOD, CAT, GR, and APX) and NEACs (PCs, TFs, carotenoids, and GSH/GSSG and ASA/DHA ratios) of tomato plants by inducing enzymes related to the secondary metabolisms of plants (GST, G6PDH, SKDH, and PAL) [8,178,179]. In vivo assays indicated that tomato leaves treated with exogenous BRs and incubated at 40 °C showed a higher increase (*p* < 0.05) in the AEs (SOD, POX, and CAT) compared to leaves placed at 25 °C [180]. Later, in vivo assays corroborated that SOD, APX, GPOD, and CAT enzymes were upregulated and OD was reduced (H_2_O_2_ and MDA) by BR application [181]. In this aspect, Zhou et al. [8] indicated that BRs induce RBOH1-NADPH oxidase activation to produce H_2_O_2_, triggering stress tolerance by the upregulation of other hormones, such as ABA.

Regarding drought and heavy metal stress (Cd and Cr), BR application improves AS and reduces OD in tomato plants when it was applied in a foliar manner [41,182,183]. In this sense, heavy metals may induce ROS, affecting DNA, proteins, and pigments, and stimulate lipid peroxidation of the cell wall. However, BR application alleviates the oxidative stress because BR upregulates GSH, AsA, and proline, which neutralize free radicals caused by heavy metals. It is noteworthy BR’s effect in tomato fruit was also beneficial for improving lyc and β-carotene due to BR accelerating the ripening process [184].

**Table 8 plants-12-03648-t008:** Effect of BRs on antioxidant system of tomato plants under abiotic stress.

Hormone (C ^1^)	Variety/Tomato Part	Age	Stress Condition	Results	Values	Reference
BRs ^2^ (100 nM)	Hezuo 903/roots	50 days old	Polychlorinated biphenyls	BRs increases AS and reduces OD	Carotenoids = 4.4–10.5%	[178]
					H_2_O_2_ = ·(13.3–20.9)%	
					O_2_^−^= −(16.5–36.0)%	
					MDA = −(7.5–8.7)%	
					SOD = 15.2–30.2%	
					POD = 64.7–152.8%	
					CAT = 15.1–20.0%	
					APX = 35.9–56.6%	
					GR = 59.0–140%	
BRs ^2^ (10.6 nM)	Amalia/leaves	21 days old	Temperature (25–40 °C)	BRs increases AEs	SOD = 58.2–81.1%	[180]
					POD = 12.1–50.5%	
					CAT = 36.2–84.9%	
BRs ^2^ (0.01–1 mg/L)	9021/leaves	55 days old	Temperature (25–40 °C) for 8 days	As increase temperature, the BRs significantly improve the AEs and OD	SOD = 12.9–13.0%	[181]
					APX = 13.0–35.7%	
					CAT = 23.4–89.2%	
					H_2_O_2_ = −(26.6–33.8)%	
					MDA = −(8.4–33.6)%	
BRs ^2^ (100 nM)	Hezuo 903/roots	50 days-old	Phenanthrene (300 µM)	Foliar application of BRs improves AS and reduces OD	PCs = 5.9%	[179]
					TF = 10.5%	
					MDA = −13.3%	
					AC(DPPH) = 15.6%	
BRs ^2^ (10^−8^ M)	K-25 and Sarvodya/leaves and fruit	60 days old and mature fruit	Cd (100 µM)	BRs improves AS (except AA)	SOD = 18.6–27.9% ^3^	[182]
					POX = 26.0–34.6% ^3^	
					CAT = 9.8–14.6% ^3^	
					Lyc = 19.5–22.1% ^4^	
					β-carotene = 8.6–14.8% ^4^	
					AA = −(15.6–19.5)% ^4^	
BRs ^2^ (10–7 M)	K-21/leaves	40 days old	Cr (10 mg/kg soil)	BRs reduces OD and increase AS	H_2_O_2_ = −50%	[141]
					MDA = −49.3%	
					EL = −28.8%	
					MG = −30.9%	
					SOD = 27.3%	
					CAT = 19.7%	
					GST = 54.5%	
					APX = 37.0%	
					GR = 48.9%	
					AA = 31.8%	
					GSH = 17.6%	
					TF = 60.6%	
BRs ^2^ (1 and 3 µM)	EC-652652 and EC-620419/leaves	67 days old	Drought	BRs reduce OD and AS	H_2_O_2_ = −(16.6–26.1)%	[183]
					SOD = 8.7–35.5%	
					Lyc = 4.1–16.0%	

^1^ Concentration; ^2^ Exogenous; ^3^ Leave; ^4^ Fruits.

## 4. Crosstalk among Hormones against Oxidative Damage Caused by Stress Factors

So far, this manuscript has described the effect of individual hormones on AS of stressed tomato plants. However, crosstalk between hormones has also been reported in studies dealing with tomato mutants altered in their hormonal pathways. Information on hormonal crosstalk resulting from responses to stressors is scarce, and valuable information is presented in Figure 1. In this sense, some controversial information was described on the hormonal effects when they interact. SA application in tomato plants under salt or nematode stress (*Ralstonia solanacearum*) significantly reduced ET biosynthesis and EL and increased AEs [110,124]. Moreover, its application significantly increased some hormones (ABA, GA_3_, IAA, and JA) and reduced the OD caused by cold temperature growth [111,185] and *R. solanacearum* [124].

On the other hand, MeJA application significantly increased ET and NEACs biosynthesis in a dose-dependent manner when it was applied against microorganisms in tomato seeds and plants [144]. Interestingly, *ERFs* genes (*ERF1*, *ERF5*, and *ERF.C4*), well-known as stress response factors associated with ET signaling, were upregulated by SA, MeJA, and ABA [80,82,90,124]. In tomato fruit, exogenous application of hormones (MeJA, BRs, and ABA) showed a positive effect on ET during the tomato fruit ripening [12,44,186]. In this aspect, the increase in ET production induced by ABA application was through the upregulation of the *LeACS2*, *LeACS4*, *LeACC1*, *LeGR*, and *LeETR6* genes [186]. Moreover, Hu et al. [187] and Vardhini and Rao [184] pointed out that BRs promote the synthesis of AC by upregulating ET biosynthesis and signaling in a dose-dependent manner. The authors theorized that AC is increased by BR regulated by the *BZR1* transcription factor, which directly regulates several genes involved in ET biosynthesis and signaling. Conversely, IAA application significantly reduced ET biosynthesis and, consequently, the ripening process, causing a reduction in lyc and α, β, and δ carotenes by downregulating the *PSY1*, *PSY3*, *PDS*, *ZlSO*, and *CrtiSO* genes and chlorophyllase 1-3 [188]. Interestingly, IAA significantly improves ABA content in tomato fruit [189]; however, this ABA increment (3–6 days) positively affects ET synthesis, as previously described [186].

Ethephon applied to tomato leaves significantly increased (2.6–10.6 times) SA content compared to control, while treatment with MeJA did not affect SA content. Interestingly, when ethephon was applied, MeJA content was reduced by decreasing *PIN2* and *PPOF* gene expression, two well-characterized wounding and insect response genes in tomato plants [90]. On the other hand, Zhou et al. [8] pointed out that BRs can induce H_2_O_2_ production in tomato plants, triggering ABA biosynthesis, which increases H_2_O_2_ production, improving stress tolerance. Thus, ABA biosynthesis was stimulated by the oxidative stress (epigallocatechin-3-gallate) in tomato plants, increasing OD and AE by the upregulation of the *NCED1* and *NCED2* genes [190]. They also indicated that, while ABA increases, GA showed a reduction, indicating an antagonist behavior between these hormones. ABA application did not present any effect on IAA when it was applied to mitigate salt stress damage. In an interesting study conducted by Heidari et al. [191], tomato seedling growth at low temperatures showed a reduction in GA_3_ and IAA in both resistant and sensitive cold tomato species, while ABA increased in both species. These results confirm the theory about the negative effect of some hormones on each other. However, there exist several interaction nodes among hormones responsible for their up- or downregulation.

## 5. Conclusions

This review shows that hormones play a pivotal role in the antioxidant response of tomato plants against biotic and abiotic stresses. Tomato plants contain different enzymatic and non-enzymatic antioxidant compounds, which can be regulated by hormones. In general, it seems that, under normal conditions, hormones are found at basal levels; however, under stress conditions, the interaction between ROS and hormones generates a loop, which increases the antioxidant system and alleviates oxidative damage. Moreover, as has been described, some hormones presented a positive, negative, or null effect among them, showing their impacts on molecular and genetic signaling. This review is valuable to clarify some important questions about hormones and their effects on oxidative damage in tomato plants. Nevertheless, further studies are needed to clarify the hormone effects on improving antioxidant responses against stress factors and how to take advantage of promoting resistance or increasing health-promoting compounds found in tomato plants.

## Figures and Tables

**Figure 1 plants-12-03648-f001:**
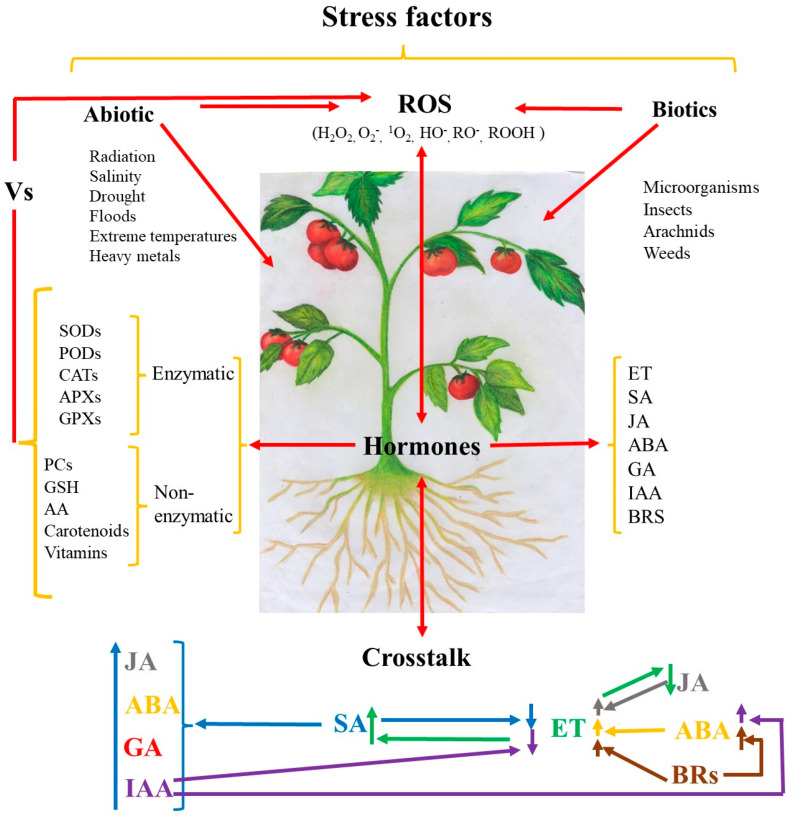
Effect of hormones and their interaction on enzymatic and non-enzymatic antioxidant systems to alleviate the biotic and abiotic stressors in tomato plants.

## Data Availability

Data is contained within the article.

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
