# Peer review of "Uncovering the Role of Hormones in Enhancing Antioxidant Defense Systems in Stressed Tomato (Solanum lycopersicum) Plants"

_plants, 2023, doi:10.3390/plants12203648_

Round 1

Reviewer 1 Report

Comments to the Authors for MS plants-2609921

The Review MS “plants-2609921” entitled “A comprehensive review about the effect of hormones on improving the antioxidant system of stressed tomato (Solanum lycopersicum) plants” by Hernández-Carranza et al., highlights the global importance of tomatoes due to their sensory and nutritional qualities. It notes that tomatoes, like other industrial crops, face challenges from various stress factors that affect their metabolic processes. Hormones play a crucial role in the tomato plants' stress responses, particularly in a complex signaling network involving antioxidants to prevent stress-related damage. Despite previous research on stress factors, hormones, and primary metabolites, there is limited knowledge about how endogenous and exogenous hormones impact secondary metabolism in tomatoes. This review aims to provide an updated overview of the role of hormones in the antioxidant system of tomato plants in response to biotic and abiotic stress factors.

In this sense, the Review is a welcome manuscript and I believe that it is worthy of publishing in the “Plants” journal. The objectives of the Review are clear, the data provided is appropriate, and the manuscript is well-written.

Minor points:

Page 3/Line 115: However, their most important role… To my opinion, it is questionable whether this is the most important role of hormones. Please rephrase.

Page 3/Line 124: …affect hormones biosynthesis and their… Rephrase to …affect the biosynthesis of hormones and their…

Page 4/Line 158: Change “fruit” to “fruits”.

Page 4/Line 174: Change “antioxidant” to “antioxidants”.

Page 6/Line 257: Change “In-plant cells” to “In plant cells”.

Page 6/Line 264: Change “encoded” to “upregulated”.

Page 7/Line 304: Change “…tomato plant…” to “…tomato plants…”.

Page 7/Line 304: Change “…initiate an ERFs biosynthesis…” to “…initiate the biosynthesis of ERFs…”.

Page 7/Line 309: Change “…informed…” to “…stated…”.

Page 7/Line 311: Change “…genes expression…” to “…gene expression…”.

Page 7/Line 312: Change “…inducing SA biosynthesis that encodes ERF1 and PR1 genes…” to “…inducing SA biosynthesis via ERF1 and PR1 upregulation…”.

Page 7/Line 318: Delete “been” and correct “Resent” to “Recent”. Change “Resent studies have been indicated…” to “Recent studies have indicated…”.

Page 8/Line 335: Change “…O2-, and H2O2…” to “…O2-, and H2O2…”.

Page 8/Line 338: Change “…temperature;…” to “…temperatures;…”.

Page 8/Line 355: Change “…temperature…” to “…temperatures…”.

Page 9/Line 406: Change “…used as stressing agent…” to “…used as a stressing agent…”.

Page 10/Line 442: Change “…of root and shoot of tomato plants…” to “…in tomato roots and shoots.”.

Page 12/Line 548: Change “…(ERF1, ERF5, and ERF.C4), a well-known stress response factors…” to “…(ERF1, ERF5, and ERF.C4), well-known as stress response factors...”.

Page 13/Line 579: Change “…temperature…” to “…temperatures…”.

Page 13/Line 582-583: “However, exist several interaction nodes among hormones responsible for up-or down-regulation between them”. Syntax error. Not clear what the Authors mean. Please rephrase.

The quality of the English Language is good. Only minor editing is required. Please refer to the "Minor points" of my review/comments.

Author Response

Thank you very much for your accurate comments to improve the quality of the manuscript. We are doing our best to improve it. The required information is written in blue in the current version of the manuscript.

Minor points:

Page 3/Line 115: However, their most important role… To my opinion, it is questionable whether this is the most important role of hormones. Please rephrase.

The reviewer is right, sentence was changed.

Page 3/Line 124: …affect hormones biosynthesis and their… Rephrase to …affect the biosynthesis of hormones and their…

Sentence was changed.

Page 4/Line 158: Change “fruit” to “fruits”.

Sentence was changed.

Page 4/Line 174: Change “antioxidant” to “antioxidants”.

Sentence was changed.

Page 6/Line 257: Change “In-plant cells” to “In plant cells”.

Sentence was changed.

Page 6/Line 264: Change “encoded” to “upregulated”.

Sentence was changed.

Page 7/Line 304: Change “…tomato plant…” to “…tomato plants…”.

Sentence was changed.

Page 7/Line 304: Change “…initiate an ERFs biosynthesis…” to “…initiate the biosynthesis of ERFs…”.

Sentence was changed.

Page 7/Line 309: Change “…informed…” to “…stated…”.

Sentence was changed.

Page 7/Line 311: Change “…genes expression…” to “…gene expression…”.

Sentence was changed.

Page 7/Line 312: Change “…inducing SA biosynthesis that encodes ERF1 and PR1 genes…” to “…inducing SA biosynthesis via ERF1 and PR1 upregulation…”.

Sentence was changed.

Page 7/Line 318: Delete “been” and correct “Resent” to “Recent”. Change “Resent studies have been indicated…” to “Recent studies have indicated…”.

Sentence was changed.

Page 8/Line 335: Change “…O2-, and H2O2…” to “…O2-, and H2O2…”.

Sentence was changed.

Page 8/Line 338: Change “…temperature;…” to “…temperatures;…”.

word was changed.

Page 8/Line 355: Change “…temperature…” to “…temperatures…”.

Sentence was changed.

Page 9/Line 406: Change “…used as stressing agent…” to “…used as a stressing agent…”.

Sentence was changed.

Page 10/Line 442: Change “…of root and shoot of tomato plants…” to “…in tomato roots and shoots.”.

Sentence was changed.

Page 12/Line 548: Change “…(ERF1, ERF5, and ERF.C4), a well-known stress response factors…” to “…(ERF1, ERF5, and ERF.C4), well-known as stress response factors...”.

Sentence was changed.

Page 13/Line 579: Change “…temperature…” to “…temperatures…”.

Sentence was changed.

Page 13/Line 582-583: “However, exist several interaction nodes among hormones responsible for up-or down-regulation between them”. Syntax error. Not clear what the Authors mean. Please rephrase.

Sentence was changed. 

Reviewer 2 Report

1  In the abstract, the authors claim that this review article offers an updated overview of both endogenous biosynthesis and exogenous hormone application in the antioxidant system of tomato plants as a response to biotic and abiotic stress factors. The paper first summarizes the enzymatic and non-enzymatic antioxidants in tomato plants followed by “Hormones and their effects on the antioxidant system of tomato plants”. In the 2nd part of the article, the authors describe 7 different types of hormones and the effects of hormones on antioxidant system of tomato plants under certain biotic and abiotic stress.  For instance, Table 2 presents the effects of salinity, cold and mycorrhiza while Table 3 only shows the abiotic stress such as salinity and temperature on antioxidant system of tomato plants. The title of Table 3 is not accurate as there are no examples of biotic stress. Same problems are found in Tables 5 and 6, which only show the effects of abiotic stress.  Furthermore, there should be clear examples of “endogenous biosynthesis and exogenous hormone application” in the antioxidant system of tomato plants as a response to biotic and abiotic stress factor.

     Other comments:

       Line 98 – “It is known that…” should read “It is well known that…”

Line 116 – “…protection against biotic and abiotic stress factors” delete “factors”.

3    Line 131- change  “aims” to “aimed”

4    Line 132- change "… submitted to” to “subjected to”

5    Line 133 to 135 “To fully achieve this purpose, the following topics are covered: (i) know the enzymatic and non-enzymatic antioxidants in tomato plants, (ii) present  and discuss the individual hormone responses on the AS of tomato plants subjected to stress factors, and (iii) know the crosstalk interaction among hormones to improve the AS of tomato plants.” – This long sentence should be re-written as 2 or 3 sentences. The word “know” has been used twice, it could be replaced with “conclude” and “understand”.

6     Line 141 “PCs are a great group of secondary metabolites” -  change “great” to “diverse”

7   Line 149 to  151 “…products, the PCs composition of tomato fruit depends on several conditions such as plant part, variety, growth, development condition, preharvest, har-150 vest, postharvest management, etc.” – change to “…products, the PCs composition of tomato fruit depends on several conditions such as plant tissues, variety, growth and development conditions, preharvest, harvest and postharvest managements, etc.”

8     Line 151 – Delete “However.”

9   Line 165 “Although carotenoids on tomato fruits…” – change to “Although carotenoids of  tomato fruits…”

  Line 311 “Helocoverpa zea” -  change to “cotton ballworm (Helocoverpa zea)”.

   Line 313 “…inoculated with Glomus clarum” – change to “…inoculated with mycorrhiza (Glomus clarum). The authors should check through the paper and provide both common names and scientific names when quoting the examples of biotic stress.

Author Response

Thank you very much for your accurate comments to improve the quality of the manuscript. We are doing our best to improve it. The required information is written in blue in the current version of the manuscript.

The reviewer can observe that in the current version of the manuscript, the endogenous biosynthesis and exogenous hormone application are differentiated.

     Other comments:

Line 98 – “It is known that…” should read “It is well known that…”

R. Change was made.

Line 116 – “…protection against biotic and abiotic stress factors” delete “factors”.

R. Change was made.

Line 131- change  “aims” to “aimed”.

R. Change was made.

Line 132- change "… submitted to” to “subjected to”

R. Change was made.

Line 133 to 135 “To fully achieve this purpose, the following topics are covered: (i) know the enzymatic and non-enzymatic antioxidants in tomato plants, (ii) present  and discuss the individual hormone responses on the AS of tomato plants subjected to stress factors, and (iii) know the crosstalk interaction among hormones to improve the AS of tomato plants.” – This long sentence should be re-written as 2 or 3 sentences. The word “know” has been used twice, it could be replaced with “conclude” and “understand”.

R. Change was made.

Line 141 “PCs are a great group of secondary metabolites” -  change “great” to “diverse”

R. Change was made.

Line 149 to  151 “…products, the PCs composition of tomato fruit depends on several conditions such as plant part, variety, growth, development condition, preharvest, har-150 vest, postharvest management, etc.” – change to “…products, the PCs composition of tomato fruit depends on several conditions such as plant tissues, variety, growth and development conditions, preharvest, harvest and postharvest managements, etc.”

R. Change was made.

Line 151 – Delete “However.”

R. Change was made.

Line 165 “Although carotenoids on tomato fruits…” – change to “Although carotenoids of  tomato fruits…”

R. Change was made.

Line 311 “Helocoverpa zea” -  change to “cotton ballworm (Helocoverpa zea)”.

R. Change was made.

Line 313 “…inoculated with Glomus clarum” – change to “…inoculated with mycorrhiza (Glomus clarum). The authors should check through the paper and provide both common names and scientific names when quoting the examples of biotic stress.

R. Change was made.

Reviewer 3 Report

In my opinion, this is still a good article with a general overview of the tomato antioxidant defense system, the effects of hormones on the antioxidant defense system, and the role of hormone interactions on antioxidants. It should be noted that the authors have used a large amount of literature to synthesize the relevant findings, and I think the article can be worthy of publication, in light of the following minor issues:

1) The title of the article does not match the topic and needs to be revised. The title of the current manuscript is too common, and I think  a good title should be used such as 'Uncovering the role of hormones in enhancing antioxidant defense systems in stressed tomato plants'

2) The abstract needs to be re-edited, in particular what role the various types of hormones related to the topic play in it is not covered, it is too general and lacks detail.

3) In the Introduction, I believe that this latest article 10.1186/s40538-022-00368-2 is able to support this review. In addition, I also think that a figure can be added in the Introduction about antioxidant defense systems. There are so many figures of mechanisms in this area that the authors can draw one, which should be very simple.

4) I think that 2.3 Hormones and their effects on the antioxidant system of tomato plants could be upgraded to 3 Hormones and their effects on the antioxidant system of tomato plants; similarly, 2.4 Crosstalk among hormones against oxidative damage caused by stress factors can be upgraded to 4 Crosstalk among hormones against oxidative damage caused by stress factors

5) The title of Figure 1 should be developed to introduce the detail of Figure 1 inside. Please re-edit the title of Figure 1. 

Author Response

Thank you very much for your accurate comments to improve the quality of the manuscript. We are doing our best to improve it. The required information is written in blue in the current version of the manuscript.

The title of the article does not match the topic and needs to be revised. The title of the current manuscript is too common, and I think  a good title should be used such as 'Uncovering the role of hormones in enhancing antioxidant defense systems in stressed tomato plants'

We agree with the reviewer's title suggestion. Therefore, it was changed.

The abstract needs to be re-edited, in particular what role the various types of hormones related to the topic play in it is not covered, it is too general and lacks detail.

Some information was added to the abstract.

In the Introduction, I believe that this latest article 10.1186/s40538-022-00368-2 is able to support this review. In addition, I also think that a figure can be added in the Introduction about antioxidant defense systems. There are so many figures of mechanisms in this area that the authors can draw one, which should be very simple.

The suggested article was added in the introduction section. Although a figure about antioxidant defense systems could be added to the manuscript, as the reviewer mentioned, this information is available elsewhere and very easy for the reader to access to this information. Moreover, the manuscript is too long on its own. Therefore, adding another figure or table increases its extension.

I think that 2.3 Hormones and their effects on the antioxidant system of tomato plants could be upgraded to 3 Hormones and their effects on the antioxidant system of tomato plants; similarly, 2.4 Crosstalk among hormones against oxidative damage caused by stress factors can be upgraded to 4 Crosstalk among hormones against oxidative damage caused by stress factors

R. Change was made.

The title of Figure 1 should be developed to introduce the detail of Figure 1 inside. Please re-edit the title of Figure 1. 

R. The title of the Figure 1 was improved.

Round 2

Reviewer 2 Report

Most concerns have been addressed.